# Mortality and Associated Causes in Hemophagocytic Lymphohistiocytosis: A Multiple-Cause-of-Death Analysis in France

**DOI:** 10.3390/jcm12041696

**Published:** 2023-02-20

**Authors:** Solène La Marle, Gaëlle Richard-Colmant, Mathieu Fauvernier, Hervé Ghesquières, Arnaud Hot, Pascal Sève, Yvan Jamilloux

**Affiliations:** 1Département de Médecine Interne, Hôpital de la Croix Rousse—Hospices Civils de Lyon, Université Claude Bernard-Lyon 1, 69002 Lyon, France; 2Département de Biostatistique-Bioinformatique, Pôle Santé Publique, Hospices Civils de Lyon, Université Claude Bernard-Lyon 1, 69000 Lyon, France; 3Département d’Hématologie, Hôpital Lyon Sud—Hospices Civils de Lyon, Université Claude Bernard-Lyon 1, 69495 Lyon, France; 4Département de Médecine Interne, Hôpital Edouard Herriot—Hospices Civils de Lyon, Université Claude Bernard-Lyon 1, 69003 Lyon, France; 5Lyon Immunopathology Federation (LIFE), Université Claude Bernard-Lyon 1, 69000 Lyon, France

**Keywords:** hemophagocytic lymphohistiocytosis, HLH, mortality, associated causes of death

## Abstract

Hemophagocytic lymphohistiocytosis (HLH) is a severe hyperinflammatory syndrome with an overall mortality rate of 40%. A multiple-cause-of-death analysis allows for the characterization of mortality and associated causes over an extended period. Death certificates, collected between 2000 and 2016 by the French Epidemiological Centre for the Medical Causes of Death (CepiDC, Inserm), containing the ICD10 codes for HLH (D76.1/2), were used to calculate HLH-related mortality rates and to compare them with the general population (observed/expected ratios, O/E). HLH was mentioned in 2072 death certificates as the underlying cause of death (UCD, *n* = 232) or as a non-underlying cause of death (NUCD, *n* = 1840). The mean age at death was 62.4 years. The age-standardized mortality rate was 1.93/million person-years and increased over the study period. When HLH was an NUCD, the most frequently associated UCDs were hematological diseases (42%), infections (39.4%), and solid tumors (10.4%). As compared to the general population, HLH decedents were more likely to have associated CMV infections or hematological diseases. The increase in mean age at death over the study period indicates progress in diagnostic and therapeutic management. This study suggests that the prognosis of HLH may be at least partially related to coexisting infections and hematological malignancies (either as causes of HLH or as complications).

## 1. Introduction

Hemophagocytic lymphohistiocytosis (HLH) is a rare hyperinflammatory syndrome characterized by the uncontrolled activation of the immune system, which can lead to life-threatening multiple organ failure. Primary HLH (pHLH) is caused by genetic mutations [1,2]. Secondary HLH (sHLH) can be caused by a variety of underlying conditions, the most common being hematological malignancies, infections, rheumatic diseases, and solid cancers [3]. Although an understanding of the mechanisms leading to HLH remains incomplete, a hallmark of HLH is a significant and uncontrolled increase in proinflammatory cytokines (IL-1, IL-6, TNF-α, and IFN-γ) and aberrant activation of cytotoxic cells [4,5,6]. A defect in antigen clearance is probably a pivotal mechanism for this inflammatory hyperactivation.

The mortality of HLH varies according to the underlying cause but is typically high, around 40% (from 2–19.5% mortality in rheumatic diseases to 72% in some lymphomas) [7,8].

Epidemiological data on HLH mortality are scarce, and most studies have been limited to specific populations or focused on pHLH mortality [9,10]. Moreover, most studies were monocentric with small sample sizes, although there are a few retrospective multicentric studies [3,11]. To gain insight into HLH-related mortality, we performed a multiple-cause-of-death (MCOD) analysis based on French death certificates, which are systematically collected by the French Epidemiological Centre for the Medical Causes of Death (CepiDC). Conducting an MCOD analysis provides comprehensive data at the national level with complete coverage of all deaths over a selected period. This method has been used previously to study mortality in rare chronic diseases [12,13]. The aims of this study is, therefore, to assess mortality rates related to HLH and associated conditions, and to compare them with those of the general population.

## 2. Materials and Methods

### 2.1. Data Source and Retrieval

CepiDC is part of the French National Institute of Health and Medical Research (Inserm). It has recorded and logged all death certificates issued in France since 1979. CepiDC provides free and anonymous datasets to researchers in France. This retrospective study was based on the death certificates registered from 2000 to 2016 (i.e., the latest complete dataset available in 2022). The use of these anonymized data does not require ethics committee approval.

In compliance with the WHO standards, French certificates consist of two parts. Part I lists data related to the death and to the patient (name, birthdate, address, place/time of death, and circumstances of death), and part II lists data related to the causes of death, anonymously. Part II is divided into two subparts: the first one lists the “diseases related to the morbid process leading to death” in reverse order of causality, with the last condition listed being defined as the underlying cause of death (UCD); the second one lists the “other significant conditions contributing to death, but not related to the disease or condition causing it”. All diseases not listed as the UCDs are considered as non-underlying causes of death (NUCDs). 

Causes of death were categorized according to the WHO International Classification of Diseases, 10th revision (ICD-10). All death certificates mentioning HLH, either as the UCD or an NUCD, were included. The corresponding codes were D76.1 (hemophagocytic lymphohistiocytosis) and D76.2 (infection-associated hemophagocytic lymphohistiocytosis). Death certificates that mentioned alternative diagnoses (e.g., Erdheim Chester disease coded as D76.1), either as the UCD or an NUCD, were excluded from analysis. HLH recorded under ICD-10 code D76.2 was classified as HLH secondary to an infection regardless of the availability of infection details on the death certificates. The death certificate data did not indicate the activity of the HLH (active, in remission, etc.). For each certificate, only two associated causes were analyzed (i.e., the NUCDs noted first on the certificate when it was not a non-specific diagnosis). These associated causes were classified into the following groups, based on the categories pre-defined by CepiDC (available at www.https://opendata-cepidc.inserm.fr/, accessed on 10 January 2023):Malignancies which were further subdivided into solid tumors and hematological malignancies; hematological malignancies were detailed (lymphomas, chronic lymphocytic leukemia, myeloid hemopathy, and a final unspecified category);Infections which were subdivided into the following subcategories: HIV; EBV; CMV; tuberculosis; bacterial causes (including all septic shocks without identified germ); and a final category encompassing fungal, parasitic, or viral causes other than those previously cited;Other categories were neurological, cardiovascular, drug-related, auto-immune, or inflammatory causes without subdivision;Other diagnoses which did not meet the definition of the previously cited causes were classified in other causes. Primary and secondary HLH were not separated because this information was not available in most cases or did not seem to have been reported appropriately. In fact, some D76.1 was reported in older patients. Thus, HLH codes were gathered and analyzed as a whole under the terminology “HLH”.

### 2.2. MCOD Analysis

A number of death certificates that listed HLH either as the UCD or an NUCD, regardless of its place on the death certificate, were considered in the MCOD analysis. 

For each death certificate that listed HLH as an NUCD, the UCDs were examined and the observed/expected ratios (O/E) were calculated for three age groups (<45 years, 45–64 years, and 65 years and over). To calculate these ratios, it was necessary to compare the observed number of deaths with the expected number of deaths under an assumption of the independence of the two causes. The expected number of deaths was calculated by multiplying the number of HLH-related deaths by the number of deaths of the corresponding cause, divided by the number of deaths of any cause. Confidence intervals were derived according to Ulm (1990).

### 2.3. Statistical Analysis

Quantitative variables were described as means and standard deviations (SDs) while qualitative variables were described as frequencies and percentages of each modality.

Observed mortality rates were calculated using the corresponding person-years from the French population (period 2000–2016) according to sex, age, and year. Age-standardized mortality rates (aSMRs; per 1 million person-years) were then calculated by a direct method, using the age distribution of the 2006 general population of France as reference. In order to consider the effects of sex, year, and age on the mortality rate, we modeled the mortality rate with a penalized Poisson regression model (generalized additive model with Poisson distribution). The non-linear effect of age was modeled using a penalized spline while the effect of year was considered as linear (after checking for non-linearity). We also considered the effect of sex and a linear interaction between year and age. The final model was:$$\log\left(\mathrm{Rate}\right)=\beta_{0}+s\;\left(\mathrm{age}\right)+\beta_{\mathrm{year}}\times \mathrm{year}+\beta_{\mathrm{male}}\times \left(\mathrm{sex}=\mathrm{men}\right)+{\;\beta }_{\mathrm{age}:\mathrm{year}}\mathrm{age}\times \mathrm{year}$$
s(age) is a restricted cubic spline containing 4 parameters. 

The coefficient β_Men was estimated at 0.575, which means that the ratio of rates between men and women is given by the quantity exp(0.575) = 1.78. Thus, the mortality rate of men is 1.78 times higher than that of women at the corresponding age and year. The implementation was performed using the R packagemgcv (v 1.8-31, Wood 2011).

For each UCD, when HLH was an NUCD, we compared the observed number of deaths (currently in the database) with the expected number (E) under an assumption of the independence of the two causes. This number was calculated from the number of deaths for which NUCD = HLH (B), the number of deaths for which UCD = cause studied (A), and the total number of deaths from any cause (T): E = (A × B)/T.

The *p*-value is interpreted, under the assumption that the true ratio is equal to 1, as the probability of obtaining a ratio at least as extreme as the one obtained from the study data.

## 3. Results

### 3.1. Characteristics of the HLH Decedents 

Between 2000 and 2016, the CepiDC recorded 9,165,205 deaths in France. Overall, 2072 death certificates reported a diagnosis of HLH, 232 (11.2%) with HLH as the UCD and 1840 (88.8%) with HLH as an NUCD. Within this population, 825 decedents were women and 1237 were men (sex ratio 1.5, Table 1).

The mean (±SD) age at death was 62.4 years (±6.3) with a progressive increase over the years (51.6 years in 2000 versus 65.7 years in 2016, Figure 1).

### 3.2. Causes of Death

In 83.2% of the death certificates mentioning HLH as the UCD, no associated cause was mentioned (Table 2). 

When HLH was an NUCD, the most frequently reported UCDs were hematological causes (42%), with lymphoma being the leading cause (26.2%) (Table 3). Infections were the second most frequently reported UCDs (39.4%), mainly caused by bacterial infections (27.2%). This was followed by solid tumors which represented 10.4% of the UCDs. 

### 3.3. Comparison with the At-Risk Population 

Compared to all-cause deaths in the general population, a greater proportion of HLH-related deaths occurred in men and women aged <64 years (Figure 2). 

The overall age-standardized mortality rate was 1.93 per million person-years. This rate increased over time, from 1.22 million person-years (for the period 2000–2004) to 2.79 per million person-years (for the period 2013–2016). These rates were consistently greater in men than in women during each period (overall male/female sex ratio at 1.84) (Table 4 and Figure 3).

The mortality rate in men decedents between 15 and 30 years was constant between 2000 and 2016 (Figure 4, left panel). However, as age increased, an upward trend appeared (the rate increased by factors of 2 for 70-year-olds and 4 for 90-year-olds). This was similar for women, even though the rate was 1.8 times lower, as shown in the same figure. Increased mortality rates were observed in decedents aged ≥50 and these increases were dependent on the study period as they were greater in the most recent years (Figure 4, right panel).

### 3.4. Observed/Expected Mortality Ratio

For most of the UCDs studied, there was a significant excess of mortality among the decedents whose death certificate mentioned HLH as an NUCD (Table 5). 

Excess mortality, as illustrated by higher O/E ratios, increased with age in decedents with HLH and HIV or CMV infection as the UCD. The greatest O/E ratios were observed in decedents with CMV infections. This was mainly seen in older decedents. 

In contrast, excess mortality was less prominent in older decedents with tuberculosis or other infectious causes mentioned as the UCDs. For decedents with hematological causes mentioned as the UCDs, excess mortality was noted regardless of the age class.

Interestingly, O/E ratios were <1 for solid tumors and cardiovascular causes. These O/E ratios varied between 0.041 and 0.29 for solid tumors, depending on the age group, and between 0.08 and 0.43 for cardiovascular causes (Table 5).

## 4. Discussion

This large MCOD analysis of HLH has shown that hematological diseases were the most frequently associated causes mentioned on death certificates, followed by infectious diseases. 

As described in the literature, hematological diseases were likely to be the triggering conditions for sHLH, with lymphoma being the most common [3]. Chronic lymphocytic leukemia (CLL) has rarely been reported in the literature as an underlying cause of sHLH, whereas it accounted for 3.2% of the death certificates mentioning sHLH in this study. A previous review of the literature showed that CLL was responsible for 1.46% of cases of HLH associated with malignancy [14]. In most cases, HLH was attributed to chemotherapy, particularly fludarabine [14,15]. It can also be difficult to distinguish CLL from Richter syndrome, either at the time of diagnosis or at the time of death. This may be one of the reasons why CLL has rarely been cited in the literature as a trigger for sHLH. In our study, when both causes were listed as associated causes, lymphoma was retained as the main cause associated with HLH. 

Infectious diseases were the second most frequent causes listed on death certificates mentioning sHLH. Regarding viral etiologies, only CMV, HIV, and EBV were distinguished. The percentage of death certificate mentioning both CMV or EBV and sHLH were consistent with French epidemiological data. Surprisingly, tuberculosis was rarely mentioned in association with sHLH, whereas it is a classical infectious trigger in the literature [16]. One could hypothesize that sHLH associated with tuberculosis is not a major driver of mortality, but this remains to be demonstrated in prospective studies.

Although studies based on death certificates do not allow definitive conclusions to be drawn, the frequencies of conditions associated with HLH were mostly consistent with those in the literature.

With regard to causes of death by age group, HIV and lymphoma were the most frequently reported causes of death in the oldest decedents (≥65 years). Hematological causes (other than lymphoma, CLL, and myeloid causes) were predominant in the youngest decedents (<65 years). This indicates that age at death is related to the underlying cause. These associations may thus suggest that (i) certain underlying conditions are associated with higher mortality (those mentioned more frequently in younger subjects) and that (ii) certain underlying causes are more likely to occur according to age category and, thus, may guide the initial etiological search.

In our study, solid tumors and HLH were mentioned in association on 10.4% of death certificates. Solid tumors are not a common cause of sHLH. The largest previous study on HLH in adults, conducted in 2197 patients, found only 1.46% of HLH secondary to solid tumors [11,13]. In clinical practice, unexplained fevers or cytopenia in the context of solid tumors are likely to be attributed to cancer or to chemotherapy; therefore, HLH may be underdiagnosed in this population [14]. Different solid tumors were found in this study; the main ones were prostatic, pulmonary, and breast cancers.

Over the study period, the average age at death increased significantly, with a difference of 14 years between 2000 and 2016. The better recognition of HLH (especially in older people) may explain this increase. Additionally, the availability of some protocols (HLH-1994, then HLH-2004), may have improved the management and the outcome of HLH, which could lead to an increase in age at death [17,18,19]. Over the same period in France, life expectancy increased from 79 to 82 years. 

Compared to all-cause deaths in the general population, a greater proportion of HLH-related deaths occurred in men and women aged <64 years, probably because HLH occurs more often in this population (mean age at diagnosis for lymphoma-associated HLH is 53 years old) and the mortality rate is still high. Moreover, we did not separate primary and secondary HLHs in this study, which can explain the difference for the group <44 years.

Age-standardized mortality rates also increased throughout the course of the study. Several hypotheses could explain these variations. First, the incidence of HLH in the general population increased over this period of time, likely due to better diagnostic performances (broadening of knowledge of this syndrome, or appearance of diagnostic scores). Second, the prevalence of the underlying diseases also increased with the aging of the general population, especially malignancies. At the same time, age-standardized mortality rates were rather stable between the ages of 15 and 40, probably because there were few changes in the incidence or the prevalence of underlying causes of HLH. Primary HLH occurred mostly before 15 years old and underlying causes such as malignancies occurred often after 40 years old. 

An analysis of the death certificates revealed a significant increase in the O/E ratio in cases of CMV infection. This suggests that mortality induced by CMV infection is strongly associated with the existence of HLH; however, it cannot be excluded that patients who died of sHLH were more systematically tested for CMV. CMV infection, and its complication by sHLH, usually occurs in immunocompromised patients, a favorable background that may have generally increased mortality in these settings. The association between HLH and HIV infection was also significant, with higher O/E ratios in older patients. It can be inferred that age (intrinsically) and duration of HIV infection (which could not be known or estimated) were factors favoring mortality [20].

Mortality for all hematological causes in sHLH was also higher than expected in the general population, with a higher O/E ratio for lymphoma and other hematological causes than for CLL and myeloid hemopathies. In the literature, hematological causes are the underlying conditions of sHLH that lead to the highest mortality rate, exceeding 80% of cases [21,22]. Our results are, therefore, consistent with these data.

Surprisingly, the O/E ratio was <1 for the association of HLH with solid tumors or cardiovascular causes. Although discussing the protecting factors involved is beyond the scope of this paper, it is interesting to note that these underlying conditions are not associated with excess mortality when HLH is associated. This may be due to collection bias, cardiovascular causes, and solid cancers being the most frequent causes of mortality in the study population (i.e., French decedents). Thus, this association to HLH may be merely coincidental, and these underlying conditions are probably not related to HLH. Such results are consistent with the literature on HLH that does not list these causes as underlying causes, at least not frequently [23,24]. This study has several limitations. Firstly, this retrospective research was based on data collected from death certificates, which may sometimes have been filled in by doctors who did not know the patient well, leading to incorrect, imprecise, or poorly prioritized causes of death. Secondly, in the CépiDC databases, diseases are coded according to ICD-10, but it is not possible to know the precise cause of death. For example, certificates indicating “sepsis” or “septic shock” do not specify the germ involved and have been categorized as “bacterial causes”. In addition, some subcategories of hematological causes were recorded with terminology such as “hematological disease” or “lymphoproliferative syndrome”. In addition, we limited the number of associated causes to two, those that seemed most relevant to us after reviewing the literature and the death certificate data. The diagnosis of HLH is often underestimated, so it is possible that some cases were not reported [25,26]. Finally, the time elapsed between the diagnosis of HLH and death is not known, nor is it indicated whether patients received treatment. Finally, it is not possible to compare data with prospective or retrospective cohorts. However, these large MCOD analyses should not be taken as definitive association or correlation studies but, rather, as big data suggestive of these associations.

## 5. Conclusions

Despite the limitations of this study, its large size underscores several messages: the age-standardized mortality rate increased over time, possibly due to better recognition of HLH. The associated causes are varied but those already known for HLH, namely infections and hematological diseases, still remain. Interestingly, CLL has been identified in association with HLH on death certificates, which is not an association often reported in the literature and deserves further investigation. 

New diagnostic tools and therapeutic strategies should modify these epidemiological data, and a longitudinal follow-up of HLH-associated mortality will surely provide evidence of these changes.

## Figures and Tables

**Figure 1 jcm-12-01696-f001:**
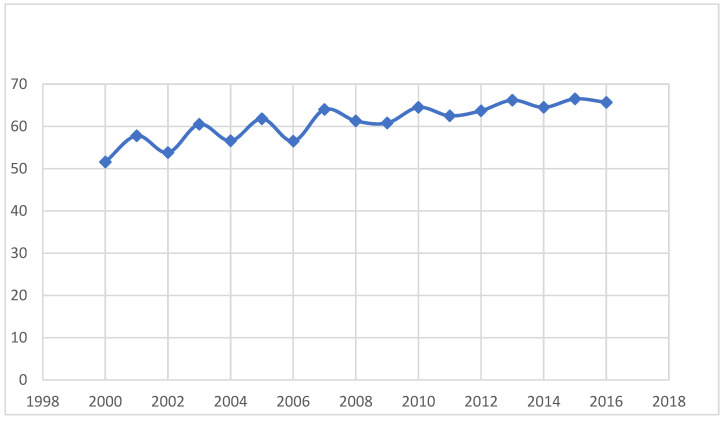
Mean age at death by years.

**Figure 2 jcm-12-01696-f002:**
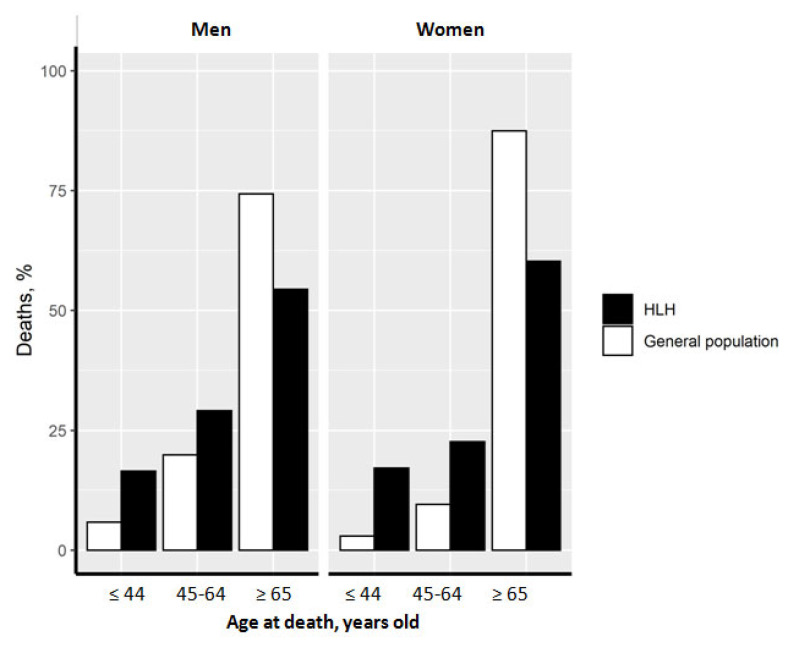
Distribution of deaths according to age. The distribution of HLH-related deaths (black bars) as well as that in the general population (white bars) according to age is presented among men and women.

**Figure 3 jcm-12-01696-f003:**
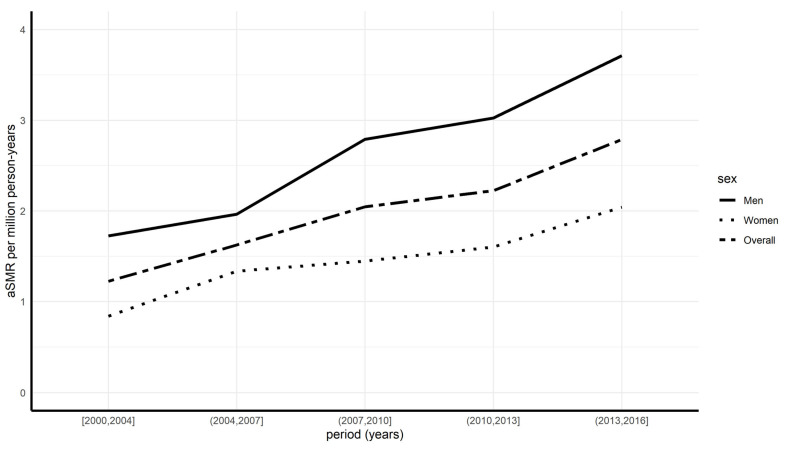
Age-standardized mortality rates by year and sex.

**Figure 4 jcm-12-01696-f004:**
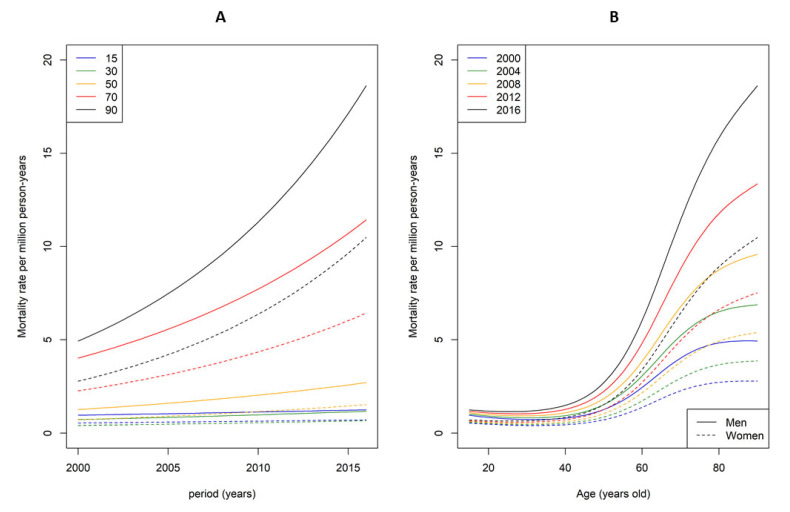
Mortality rates predicted by the flexible model (as a function of year for multiple ages (**A**) and as a function of age for multiple years (**B**)). Men are represented by solid lines and women by dashed lines.

**Table 1 jcm-12-01696-t001:** Description of the population according to the main cause of death.

Year	(2000, 2004)	(2004, 2007)	(2007, 2010)	(2010, 2013)	(2013, 2016)	Total
	*n* = 357	*n* = 300	*n* = 391	*n* = 442	*n* = 582	*n* = 2072
Sex						
HLH = UCD						
*n* = 232 (11.2%)						
Female	30	19	19	27	23	118
Male	22	18	19	26	29	114
	52	37	38	53	52	232
HLH = NUCD						
*n* = 1840 (88.8%)						
Female	103	119	134	148	213	717
Male	202	144	219	241	317	1123
	305	263	353	389	530	1840
Age						
HLH = UCD						
*n* = 232 (11.2%)						
≤44	16	9	7	9	9	50
45–64	11	5	6	6	8	36
≥65	25	23	25	38	35	146
	52	37	38	53	52	232
HLH = NUCD						
*n* = 1840 (88.8%)						
≤44	81	53	53	53	57	297
45–64	79	66	116	106	146	513
≥65	145	144	184	230	327	1030
	305	263	353	389	530	1840

UCD: underlying cause of death; NUCD: non-underlying cause of death, HLH: Hemophagocytic lymphohistiocytosis.

**Table 2 jcm-12-01696-t002:** NUCD when HLH was the UCD.

Sex	Women	Men	Total
	*n* = 118	*n* = 114	*n* = 232
NUCD			
Myeloid hemopathy	1 (0.8%)	0 (0.0%)	1 (0.4%)
Other hematological causes	2 (1.7%)	1 (0.9%)	3 (1.3%)
Bacterial causes	1 (0.8%)	0 (0.0%)	1 (0.4%)
Auto-immune causes	2 (1.7%)	7 (6.1%)	9 (3.9%)
Cardiovascular causes	5 (4.2%)	7 (6.1%)	12 (5.2%)
Neurological causes	0 (0.0%)	1 (0.9%)	1 (0.4%)
Other causes	7 (5.9%)	5 (4.4%)	12 (5.2%)
No associated	100 (84.8%)	93 (81.6%)	193 (83.2%)

NUCD: non-underlying cause of death.

**Table 3 jcm-12-01696-t003:** Description of the UCDs when HLH was an NUCD.

Sex	Women	Men	Total
	*n* = 717	*n* = 1123	*n* = 1840
UCD			
Solid tumors	83 (11.6%)	109 (9.7%)	192 (10.4%)
Hematological causes			
Lymphomas	198 (27.6%)	284 (25.3%)	482 (26.2%)
Chronic lymphocytic leukemia	16 (2.2%)	42 (3.7%)	58 (3.2%)
Myeloid hemopathy	44 (6.1%)	70 (6.2%)	114 (6.2%)
Other hematological causes	41 (5.7%)	77 (6.9%)	118 (6.4%)
Infectious causes			
Bacterial causes	184 (25.7%)	316 (28.1%)	500 (27.2%)
CMV	14 (2.0%)	9 (0.8%)	23 (1.2%)
EBV	20 (2.8%)	21 (1.9%)	41 (2.2%)
HIV	9 (1.3%)	53 (4.7%)	62 (3.4%)
Tuberculosis	9 (1.3%)	17 (1.5%)	26 (1.4%)
Other infectious causes	31 (4.3%)	42 (3.7%)	73 (4.0%)
Auto-immune causes	18 (2.5%)	13 (1.2%)	35 (1.7%)
Cardiovascular causes	19 (2.6%)	25 (2.2%)	44 (2.4%)
Neurological causes	18 (2.5%)	22 (2.0%)	40 (2.2%)
Drug-related causes	1 (0.1%)	2 (0.2%)	3 (0.2%)
Other causes	12 (1.7%)	21 (1.9%)	33 (1.8%)

CMV: cytomegalovirus; EBV: Epstein–Barr virus; HIV: human immunodeficiency virus; UCD: underlying cause of death; NUCD: non-underlying cause of death.

**Table 4 jcm-12-01696-t004:** Age-standardized mortality rates per 1 million people years.

	(2000, 2004)	(2004, 2007)	(2007, 2010)	(2010, 2013)	(2013, 2016)	Total
Overall	1.22	1.62	2.04	2.22	2.79	1.93
Male	1.72	1.96	2.79	3.02	3.71	2.6
Female	0.84	1.34	1.45	1.6	2.04	1.41
Male/female ratio	2.06	1.47	1.93	1.89	1.82	1.84

**Table 5 jcm-12-01696-t005:** Observed/expected ratio by major causes of death (95% confidence intervals are given in parentheses).

Age Class	Age <45		Age: 45–64		Age >65	
	O/E Ratio (95% CI)	*p*	O/E Ratio (95% CI)	*p*	O/E Ratio (95% CI)	*p*
Solid tumors	0.22 (0.11–0.38)	<0.001	0.29 (0.22–0.36)	<0.001	0.041 (0.33–0.49)	<0.001
Hematological causes						
Lymphomas	22.35 (16.93–28.96)	0.015	25.26 (21.29–30.43)	<0.001	32.56 (28.92–36.47)	<0.001
Chronic lymphocytic leukemia	NA	NA	16.26 (7.8–29.91)	<0.001	19.68 (14.51–26.09)	<0.001
Myeloid hemopathy	4.6 (2.73–7.28)	<0.001	5.45 (3.53–8.05)	<0.001	6.15 (4.81–7.76)	<0.001
Other hematological causes	74.84 (40.92–125.57)	<0.001	16.11 (10.94–22.86)	<0.001	39.01 (30.58–49.05)	<0.001
Infectious causes						
CMV	166.7 (54.14–389.12)	<0.001	415.36 (199.18–763.85)	<0.001	436.62 (188.5–860.32)	<0.001
HIV	6.69 (4.29–9.95)	<0.001	16.28 (10.9–23.38)	<0.001	54.86 (25.08–104.14)	<0.001
Tuberculosis	28.58 (10.49–62.22)	<0.001	28.15 (14.05–50.37)	<0.001	6.2 (2.84–11.77)	<0.001
Other infectious diseases	29.84 (24.7–35.74)	<0.001	23.89 (20.34–27.88)	<0.001	15.06 (13.49–16.76)	<0.001
Cardiovascular causes	0.43 (0.21–0.8)	<0.001	0.12 (0.05–0.23)	<0.001	0.08 (0.05–0.12)	<0.001
Other causes	0.16 (0.11–0.23)	<0.001	0.18 (0.12–0.26)	<0.001	0.11 (0.08–0.15)	<0.001

NA: not applicable; CMV: cytomegalovirus; HIV: human immunodeficiency virus.

## Data Availability

All data are shown in the manuscript.

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
