# Peer review of "Mortality and Associated Causes in Hemophagocytic Lymphohistiocytosis: A Multiple-Cause-of-Death Analysis in France"

_jcm, 2023, doi:10.3390/jcm12041696_

Round 1
Reviewer 1 Report (Previous Reviewer 2)
All revisions were done sufficiently.
Author Response
Authors thank the Reviewer for his/her comments.
1) It would be better if add "in France" to the title, because the etiology of sHLH in west and Asia is different, and this study can only explain the situation in France at that period.
Author's response >> We have added "in France" in the title.
2) The author mentioned that “Primary and secondary HLH were not separated because this information was not available in most cases, or seemed not appropriately reported”, which may under estimate the death mortality of primary HLH overall. It would be better if the author clarified this situation in the abstract, too. In addition, if possible, adding a group under 14 years in the age groups analysis may better highlight the death mortality of primary HLH.
Author's response >> Primary HLH is a very rare condition that is not well coded in ICD-10 and sometimes D76.1 and D76.2 seem to be used incorrectly by the coder/coding centre. For example, we had several D76.1 codes (normally to be used for primary/family HLH) in older people that really suggested coding errors. Thus, this information was only available when it was handwritten and reported by the CepiDC (i.e. in an extremely small number of cases that could not allow conclusions to be drawn). However, the authors believe that the codes D76.1/2 were sometimes used one for the other and preferred to combine the two codes without analysing them separately and to speak of HLH as a whole (without differentiating primary from secondary). We added (page 3, line 106) : "Primary and secondary HLH were not separated because this information was not available in most cases, or seemed not appropriately reported. Actually, some D76.1 were reported in older patients. HLH codes were thus gathered and analysed as a whole under the terminology “HLH”."
For the same reasons, looking at younger categories would not yield sufficient data for analyses, since secondary HLH can occur before 14 (and some fHLH may also have been coded D76.2).
3)“When HLH was an NUCD, the most frequently associated UCD were haematological diseases 29 (42%), infections (39.4%) and solid tumors (10.4%).” What was the state of HLH in these patients at the time of death? Remission or progress?
Author's response >> The few data from death certificates do not allow such a degree of analysis, only the presence/absence of HLH (as a code D76.1 or D76.2) is mentioned on the certificates. We added (page3; line 89) : "The death certificate data do not indicate the activity of the HLH (active, in remission, etc.)."
4)The author states “As compared to the general population, HLH decedents were more likely to have associated CMV infections or hematological diseases.” What dose general population refer to? Please show this comparison in the result section.
Author's response >> This is described in the Method section; this analysis is the O/E (observed/expected ratio) and compared the frequency of each NUCD associated with HLH (when HLH was the UCD) to its frequency in the French decedents (ie. in the general population) over the same period (page 3 l 112-115, page 4, lines 124-127). The results of this analyses are shown in Table 5.
5) In the abstract, the author states “this study suggests that the prognosis of HLH is mainly related to coexisting infections and haematological malignancies.” As the prognosis of HLH were related to the etiology, treatment, complications and many other factors. It is impossible to draw the definitive conclusions in this study.
Author's response >> Actually, the study shows HLH deaths was more frequently associated with coexisting infections and haematological malignancies (either as causes or as complications) in France between 2000-2016. The opening sentence is a suggestion the prognosis may be related to such factors. Since the conclusions seemed too definitive to the Reviewer, the authors have changed it for "This study suggests that the prognosis of HLH may be at least partially related to coexisting infections and haematological malignancies (either as causes of HLH or as complications)."
Reviewer 2 Report (Previous Reviewer 1)
The author collected 2072 death certificates which related to HLH between 2000 and 2016 from CepiDC, and aimed to calculate HLH-related mortality rates and compare with the general population. There are some concerns as follows.
1) It would be better if add "in France" to the title, because the etiology of sHLH in west and Asia is different, and this study can only explain the situation in France at that period.
2) The author mentioned that “Primary and secondary HLH were not separated because this information was not available in most cases, or seemed not appropriately reported”, which may under estimate the death mortality of primary HLH overall. It would be better if the author clarified this situation in the abstract, too. In addition, if possible, adding a group under 14 years in the age groups analysis may better highlight the death mortality of primary HLH.
3)“When HLH was an NUCD, the most frequently associated UCD were haematological diseases 29 (42%), infections (39.4%) and solid tumors (10.4%).” What was the state of HLH in these patients at the time of death? Remission or progress?
4)The author states “As compared to the general population, HLH decedents were more likely to have associated CMV infections or hematological diseases.” What dose general population refer to? Please show this comparison in the result section.
5)In the abstract, the author states “this study suggests that the prognosis of HLH is mainly related to coexisting infections and haematological malignancies.” As the prognosis of HLH were related to the etiology, treatment, complications and many other factors. It is impossible to draw the definitive conclusions in this study.
Author Response
Authors thank the Reviewer for his/her comments.
1) It would be better if add "in France" to the title, because the etiology of sHLH in west and Asia is different, and this study can only explain the situation in France at that period.
Author's response >> We have added "in France" in the title.
2) The author mentioned that “Primary and secondary HLH were not separated because this information was not available in most cases, or seemed not appropriately reported”, which may under estimate the death mortality of primary HLH overall. It would be better if the author clarified this situation in the abstract, too. In addition, if possible, adding a group under 14 years in the age groups analysis may better highlight the death mortality of primary HLH.
Author's response >> Primary HLH is a very rare condition that is not well coded in ICD-10 and sometimes D76.1 and D76.2 seem to be used incorrectly by the coder/coding centre. For example, we had several D76.1 codes (normally to be used for primary/family HLH) in older people that really suggested coding errors. Thus, this information was only available when it was handwritten and reported by the CepiDC (i.e. in an extremely small number of cases that could not allow conclusions to be drawn). However, the authors believe that the codes D76.1/2 were sometimes used one for the other and preferred to combine the two codes without analysing them separately and to speak of HLH as a whole (without differentiating primary from secondary). We added (page 3, line 106) : "Primary and secondary HLH were not separated because this information was not available in most cases, or seemed not appropriately reported. Actually, some D76.1 were reported in older patients. HLH codes were thus gathered and analysed as a whole under the terminology “HLH”."
For the same reasons, looking at younger categories would not yield sufficient data for analyses, since secondary HLH can occur before 14 (and some fHLH may also have been coded D76.2).
3)“When HLH was an NUCD, the most frequently associated UCD were haematological diseases 29 (42%), infections (39.4%) and solid tumors (10.4%).” What was the state of HLH in these patients at the time of death? Remission or progress?
Author's response >> The few data from death certificates do not allow such a degree of analysis, only the presence/absence of HLH (as a code D76.1 or D76.2) is mentioned on the certificates. We added (page3; line 89) : "The death certificate data do not indicate the activity of the HLH (active, in remission, etc.)."
4)The author states “As compared to the general population, HLH decedents were more likely to have associated CMV infections or hematological diseases.” What dose general population refer to? Please show this comparison in the result section.
Author's response >> This is described in the Method section; this analysis is the O/E (observed/expected ratio) and compared the frequency of each NUCD associated with HLH (when HLH was the UCD) to its frequency in the French decedents (ie. in the general population) over the same period (page 3 l 112-115, page 4, lines 124-127). The results of this analyses are shown in Table 5.
5) In the abstract, the author states “this study suggests that the prognosis of HLH is mainly related to coexisting infections and haematological malignancies.” As the prognosis of HLH were related to the etiology, treatment, complications and many other factors. It is impossible to draw the definitive conclusions in this study.
Author's response >> Actually, the study shows HLH deaths was more frequently associated with coexisting infections and haematological malignancies (either as causes or as complications) in France between 2000-2016. The opening sentence is a suggestion the prognosis may be related to such factors. Since the conclusions seemed too definitive to the Reviewer, the authors have changed it for "This study suggests that the prognosis of HLH may be at least partially related to coexisting infections and haematological malignancies (either as causes of HLH or as complications)."
Round 2
Reviewer 2 Report (Previous Reviewer 1)
The authors have addressed the comments accordingly.
This manuscript is a resubmission of an earlier submission. The following is a list of the peer review reports and author responses from that submission.
Round 1
Reviewer 1 Report
In this paper, mortality and associated cause in HLH were analyzed by death certificates.Although the study suffers from limitations, the results of its large scale of cases contribute to a better understanding of HLH.
1 What was the underlying causes of HLH in the cases?When HLH was the UCD.
2 Compared to all-cause death in the general population,a greater proportion of HLH-related death occurred in men and women aged <64 years.Please analyze why HLH related death is more likely to occur in younger cases?
3 The mortality rate between 15-30 years was constant.However, as age increased, an upward trend appears. In the discussion section, the reason may be the prevalence of primary HLH was not modified.Were there any cases of primary HLH in the study?Please provide the specific type and age?
4 In the study, solid tumors and HLH were mentioned in association on 10.4% of death certificates.Please provide the specific type of solid tumor.
Reviewer 2 Report
La Marle et al. describe an epidemiological study an Hemophagocytic Lymphohistiocytosis. They observed – besides some descriptive results – two interesting findings: 1) The age-26 standardized mortality rate was 1.93/million person-years and increased over the study period. 2) An increase in mean age at death over the study period.
The study is well done, the topic of interest and adds new findings to the field. However, some points need clarification:
- Introduction: „Moreover, most studies were monocentric with small sample sizes [3].“ There are systematic reviews available, which analyzed mortality in a multicentric approach
- Methods: Punctuation is missing within the ICD-10 codes
- It is unclear whether cases were only secondary or also primary HLH.
- Please include an ethics statement.
- Please specify „Confidence intervals were derived according to Ulm (1990).“
- The statisctical analysis part is hard to understand for a non-statistician and should be rewritten. An example is „This number was calculated from the number of deaths for which NUCD = HLH (B), the number of deaths for which UCD = cause studied (A) and the total number of deaths any cause (T): E = (AxB)/T.“
- Table 1 and Figure 2: The term „44-“ may probaly not be clear to all readers.
- Is it possible to specify „solid tumors“?
- Figure 2: A and B could be replaced by men and women
- Gender should be replaced by sex
- Figure 4: B and C are not clear
- Table 5: It is unclear where the p values refer to and how the confidence intervals were calculated
- There are some typos within the manuscript, please check (e.g. line 282 „also also“).
- Discussion: The average age at death increased by 14 years. This may be rather by increased diagnosis (especially in older age) and less likely by improved therapy (which would unlikely have lead to HLH deaths). Please adjust the respective part (e.g. „[…] availability of some protocols (HLH-1994, then HLH-2004)), may have improved the management and the outcome oh HLH [16–18].“ Also, the latter point contradicts „Age-standardized mortality rates also also increased between the beginning and the end of the study.“
- Furthermore, this point is an important result: „Age-standardized mortality rates also also increased between the beginning and the end of the study.“ However, your hypotheses cannot fully explain the increased mortality. At least the thoughts behind these need better explanation.
- The sentence „pHLH (which predominates in children and young adults)“ is misleading as pHLH is a disease of early childhood and true pHLH is unlikely in adults.
- Please add a new line for the limitations
- The sentence „The diagnosis of HLH is often underestimated, so it is possible that some cases were not reported.“ deserves a reference.